# Fast Multi-Resolution Transformer Fine-tuning for Extreme Multi-label Text Classification

**Jiong Zhang**
Amazon
jiongz@amazon.com

**Wei-cheng Chang**
Amazon
chanweic@amazon.com

**Hsiang-fu Yu**
Amazon
rofu.yu@gmail.com

**Inderjit S. Dhillon**
UT Austin & Amazon
inderjit@cs.utexas.edu

## Abstract

Extreme multi-label text classification (XMC) seeks to find relevant labels from an extreme large label collection for a given text input. Many real-world applications can be formulated as XMC problems, such as recommendation systems, document tagging and semantic search. Recently, transformer based XMC methods, such as X-Transformer and LightXML, have shown significant improvement over other XMC methods. Despite leveraging pre-trained transformer models for text representation, the fine-tuning procedure of transformer models on large label space still has lengthy computational time even with powerful GPUs. In this paper, we propose a novel *recursive* approach, XR-Transformer to accelerate the procedure through recursively fine-tuning transformer models on a series of multi-resolution objectives related to the original XMC objective function. Empirical results show that XR-Transformer takes significantly less training time compared to other transformer-based XMC models while yielding better state-of-the-art results. In particular, on the public Amazon-3M dataset with 3 million labels, XR-Transformer is not only 20x faster than X-Transformer but also improves the Precision@1 from $51\%$ to $54\%$. Our code is publicly available at https://github.com/amzn/pecos.

## 1 Introduction

Many real-world applications such as open-domain question answering [1, 2], e-commerce dynamic search advertising [3, 4], and semantic matching [5], can be formulated as an extreme multi-label text classification (XMC) problem: given a text input, predict relevant labels from an enormous label collection of size $L$. In these applications, $L$ ranges from tens of thousands to millions, which makes it very challenging to design XMC models that are both accurate and efficient to train. Recent works such as Parabel [3], Bonsai [6], XR-Linear [7] and AttentionXML [8], exploit the correlations among the labels to generate label partitions or hierarchical label trees (HLTs) which can be used to shortlist candidate labels to be considered during training and inference. While these methods are scalable in terms of the size of the label collection, most of them rely only on statistical representations (such as bag-of-words) or pooling from pre-generated token embeddings (such as word2vec) to vectorize text inputs.

In light of the recent success of deep pretrained transformers models such as BERT [9], XLNet [10] and RoBerta [11] in various NLP applications, X-Transformer [12] and LightXML [13] propose to fine-tune pre-trained transformer models on XMC tasks to obtain new state-of-the-art results over the aforementioned approaches. Although transformers are able to better capture semantic meaning of textual inputs than statistical representations, text truncation is often needed in practice to reduce GPU memory footprint and maintain model efficiency. For example, X-Transformer truncates input texts

35th Conference on Neural Information Processing Systems (NeurIPS 2021).

to contain the first 128 tokens before feeding it into transformer models. Efficiency of transformer fine-tuning poses another challenge for XMC applications. Directly fine-tuning transformer models on the original XMC task with a very large label collection is infeasible as both the training time and the memory consumption are linear in $L$. In order to alleviate this, both X-Transformer and LightXML adopt a similar approach to group $L$ labels into $K$ clusters of roughly equal size denoted by $B$ and fine-tune transformers on the task to identify relevant label clusters (instead of labels themselves). If $B \approx \sqrt{L}$ and $K \approx \sqrt{L}$, then both the training time and the memory requirement of the fine-tuning can be reduced to $O(\sqrt{L})$ from $O(L)$. However, as pointed out in [8], the model performance would deteriorate due to the information loss from label aggregation. Thus, both X-Transformer and LightXML still choose a small constant $B$ ( $\leq 100$) as the size of the label clusters. As a result, transformers are still fine-tuned on a task with around $L/100$ clusters, which leads to a much longer training time compared with non-transformer based models. For example, it takes X-Transformer 23 and 25 days respectively to train on Amazon-3M and Wiki-500K even with 8 Nvidia V100 GPUs.

To address these issues, we propose XR-Transformer, an XMC architecture that leverages pre-trained transformer models and has much smaller training cost compared to other transformer-based XMC methods. Motivated by the multi-resolution learning in image generation [14–16] and curriculum learning [17], we formulate the XMC problem as a series of sub-problems with multi-resolution label signals and recursively fine-tune the pre-trained transformer on the coarse-to-fine objectives. In this paper, our contributions are as follows:

- We propose XR-Transformer, a transformer based framework for extreme multi-label text classification where the pre-trained transformer is recursively fine-tuned on a series of easy-to-hard training objectives defined by a hierarchical label tree. This allows the transformers to be quickly fine-tuned for a XMC problem with a very large number label collection progressively.
- To get better text representation and mitigate the information loss in text truncation for transformers, we take into account statistical text features in addition to the transformer text embeddings in our model. Also, a cost sensitive learning scheme by label aggregation is proposed to introduce richer information on the coarsified labels.
- We conduct experiments on 6 public XMC benchmarking datasets and our model takes significantly lower training time compared to other transformer-based XMC models to yield better state-of-the-art results. For example, we improve the state-of-the-art Prec@1 result on Amazon-3M established by X-Transformer from 51.20% to 54.04% while reducing the required training time from 23 days to 29 hours using the same hardware.

## 2   Related Works

**Sparse Linear Models with Partitioning Techniques.**   Conventional XMC methods consider fixed input representations such as sparse TF-IDF features and study different partitioning techniques or surrogate loss functions on the large output spaces to reduce complexity. For example, sparse linear one-versus-all (OVA) methods such as DiSMEC [18], PPD-Sparse [19, 20], ProXML [21] explore parallelism to solve OVA losses and reduce the model size by weight truncations.

The inference time complexity of OVA models is linear in the output space, which can be greatly improved by partitioning methods or approximate nearest neighbor (ANN) indexing on the label spaces. Initial works on tree-based methods [22, 23] reduce the OVA problem to one-versus-some (OVS) with logarithmic depth trees. Down that path, recent works on sparse linear models including Parabel [3], eXtremeText [24], Bonsai [6], XReg [25], NAPKINXC [26, 27] and XR-Linear [7] partition labels with $B$-array hierarchical label trees (HLT), leading to inference time complexity that is logarithmic in the output space. On the other hand, low-dimensional embedding-based models often leverage ANN methods to speed up the inference procedure. For example, AnnexML [28] and SLICE [29] consider graph-based methods such as HNSW [30] while GLaS [31] considers product quantization variants such as ScaNN [32].

**Shallow Embedding-based Methods.**   Neural-based XMC models employ various network architectures to learn semantic embeddings of the input text. XML-CNN [33] applies one-dimensional CNN on the input sequence and use the BCE loss without sampling, which is not scalable to XMC problems. AttentionXML [8] employs BiLSTMs and label-aware attention as scoring functions. For better scalability to large output spaces, only a small number of positive and hard negative labels

are used in model GPU training. Shallow embedding-based methods [34–38] use word embedding lookup followed by shallow MLP layers to obtain input embeddings. For instance, MACH [34] learns MLP layers on several smaller XMC sub-problems induced by hashing tricks on the large label space. Similarly, DeepXML [35] and its variant (i.e., DECAF [36], GalaXC [37], ECLARE [38]) pre-train MLP encoders on XMC sub-problems induced by label clusters. They freeze the pre-trained word embedding and learn another MLP layer followed by a linear ranker with sampled hard negative labels from HNSW [30]. Importantly, shallow embedding-based methods only show competitive performance on short-text XMC problems where the number of input tokens is small [34, 35].

**Deep Transformer Models.**   Recently, pre-trained Transformer models [9–11] have been applied to XMC problems with promising results [12, 13, 39]. X-Transformer [12] considers a two-stage approach where the first stage transformer-based encoders are learned on XMC sub-problems induced by balanced label clusters, and the second stage sparse TF-IDF is combined with the learned neural embeddings as the input to linear OVA models. APLC-XLNet [39] fine-tunes XLNet encoder on adaptive imbalanced label clusters based on label frequency similar to Adaptive Softmax [40]. LightXML [13] fine-tunes Transformer encoders with the OVA loss function end-to-end via dynamic negative sampling from the matching network trained on label cluster signals. Nonetheless, Transformer-based XMC models have larger model size and require longer training time, which hinders its practical usage on different downstream XMC problems.

## 3   Background Material

We assume we are given a training set $\{\mathbf{x}_i, \mathbf{y}_i\}_{i=1}^N$ where $\mathbf{x}_i \in \mathcal{D}$ is the $i$th input document and $\mathbf{y}_i \in \{0, 1\}^L$ is the one hot label vector with $y_{i,\ell} = 1$ indicating that label $\ell$ is relevant to instance $i$. The goal of eXtreme Multi-label Text Classification (XMC) is to learn a function $f : \mathcal{D} \times [L] \mapsto \mathbb{R}$, such that $f(\mathbf{x}, \ell)$ denotes the relevance between the input $\mathbf{x}$ and the label $\ell$. In practice, labels with the largest $k$ values are retrieved as the predicted relevant labels for a given input $\mathbf{x}$. The most straightforward model is one-versus-all (OVA) model:

$$f(\mathbf{x}, \ell) = \mathbf{w}_\ell^\top \Phi(\mathbf{x}); \ \ell \in [L], \tag{1}$$

where $\mathbf{W} = [\mathbf{w}_1, \ldots, \mathbf{w}_L] \in \mathbb{R}^{d \times L}$ are the weight vectors and $\Phi : \mathcal{D} \mapsto \mathbb{R}^d$ is the text vectorizer that maps $\mathbf{x}$ to $d$-dimensional feature vector. $\Phi(\cdot)$ could be a deterministic text vectorizer, such as the bag-of-words (BOW) model or Term Frequency-Inverse Document Frequency (TFIDF) model, or a vectorizer with learnable parameters. With the recent development in deep learning, using pre-trained transformer as the text vectorizer has shown promising results in many XMC applications [12, 13, 39]. When $L$ is large, however, training and inference of OVA model without sampling would be prohibitive due to the $O(L)$ time complexity.

To handle the extremely large output space, recent approaches partition the label space to shortlist the labels considered during training and inference. In particular, [7, 12, 13, 34, 35, 39] follow a three stage framework: partitioning, shortlisting, and ranking. First, label features are constructed to group labels into $K$ clusters $\mathbf{C} \in \{0, 1\}^{L \times K}$ where $C_{\ell,k} = 1$ denotes that label $\ell$ is included in the $k$-th cluster. Then a shortlisting model is learned to match input $\mathbf{x}$ to relevant clusters in an OVA setting:

$$g(\mathbf{x}, k) = \hat{\mathbf{w}}_k^\top \Phi_g(\mathbf{x}); \ k \in [K]. \tag{2}$$

Finally, a classification model with output size $L$ is trained on the shortlisted labels:

$$f(\mathbf{x}, \ell) = \mathbf{w}_\ell^\top \Phi(\mathbf{x}); \ \ell \in S_g(\mathbf{x}), \tag{3}$$

where $S_g(\mathbf{x}) \subset [L]$ is the label set shortlisted by $g(\mathbf{x}, \cdot)$. In the extreme case where only one label cluster is determined to be relevant to a input $\mathbf{x}$, the training and inference cost on $\mathbf{x}$ would be $O(K + \frac{L}{K})$, which in the best case scenario is $O(\sqrt{L})$ when $K = \sqrt{L}$.

For transformer based methods, the dominant time is the evaluation of $\Phi(\mathbf{x})$. But $K$ being too big or too small could still be problematic. Empirical results show that the model performance deteriorates when clusters are too big [8]. This is because that the signals coming from $B$ labels within the same cluster will be aggregated and not distinguishable, where $B$ is the cluster size. Therefore, $B$ cannot be too big to ensure a reasonable label resolution for fine-tuning. Also, as pointed out in [12], fine-tuning transformer models on large output spaces can be prohibitive. As a result, the label clusters need to

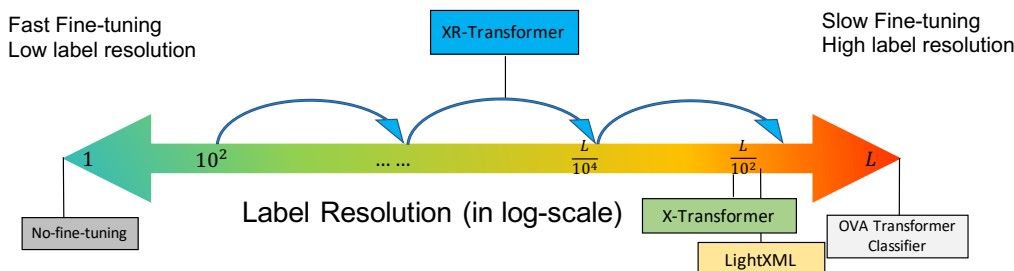

Figure 1: Illustration of fine-tuning on XMC tasks of different label resolutions. For an XMC task with a low label resolution, fine-tuning can be fast but model performance might deteriorate due to large deviation from the original XMC task. In practice, X-Transformer and LightXML adopt a XMC task with a relatively higher label resolution to ensure reasonable model performance at the cost of longer training time. The proposed XR-Transformer leverages multi-resolution learning and model bootstrapping that achieves both fast fine-tuning and good model performance.

be constructed in a way to balance the model performance and fine-tuning efficiency. In practice, both the transformer-based XMC models, such as X-Transformer and LightXML, adopt a small fixed constant as the cluster size $B (\leq 100)$, which means that training the shortlisting model $g(\mathbf{x}, k)$ is still very time consuming as the number of clusters $K \approx L/B$.

## 4 Proposed Method: XR-Transformer

As noted above, the shortlisting problem (2) is itself an XMC problem with slightly smaller output size $\frac{L}{B}$ where $B$ is the cluster size. In XR-Transformer, we apply the same three stage framework recursively on the shortlisting problem until a reasonably small output size is reached $\frac{L}{B^D}$. We can therefore follow the curriculum learning scheme and fine-tune the pre-trained transformers progressively on the sub-XMC problems with increasing output space $\{\frac{L}{B^D}, \frac{L}{B^{D-1}}, \ldots\}$. At each fine-tuning task, the candidate label set is shortlisted by the final model at the previous task. The recursive shortlisting ensures that for any input, the number of candidate labels to include in training and inference is $O(B)$ and therefore the total number of considered labels is $O(B \log_B(L))$. Also, we leverage the multi-step fine-tuning and use the embedding generated at the previous task to bootstrap the non pre-trained part for the current task. We now describe the model design in detail.

**Hierarchical Label Tree (HLT).** Recursively generating label clusters $D$ times is equivalent to building a HLT [41] of depth $D$. We first construct label features $\mathbf{Z} \in \mathbb{R}^{L \times \hat{d}}$. This could be done by applying text vectorizers on label text or from Positive Instance Feature Aggregation (PIFA):

$$\mathbf{Z}_\ell = \frac{\mathbf{v}_\ell}{\|\mathbf{v}_\ell\|}; \text{where } \mathbf{v}_\ell = \sum_{i:y_{i,\ell}=1} \Phi(\mathbf{x}_i), \forall \ell \in [L], \tag{4}$$

where $\Phi : \mathcal{D} \mapsto \mathbb{R}^d$ it the text vectorizer. Then we follow similar procedures as [8] and [13] and use balanced k-means ($k = B$) to recursively partition label sets and generate the HLT in a top-down fashion. The HLT is represented with a series of indexing matrices $\{\mathbf{C}^{(t)}\}_{t=1}^D$, such that $\mathbf{C}^{(t)} \in \{0,1\}^{K_t \times K_{t-1}}$ where $K_0 = 1$ and $K_D = L$. Equivalently, once $\mathbf{C}^{(D)}$ is constructed, the HLT can be built from bottom up through joining $B$ adjacent clusters together.

**Multi-resolution Output Space.** Multi-resolution learning has been explored in different contexts such as computer vision [14, 42]. For instance, using an output scheme with coarse-to-fine resolutions results in better image quality for generative adversarial networks [15, 16]. As an another example in meta learning, [43] learns multiclass models via auxiliary meta classes by collapsing existing classes. Nevertheless, multi-resolution learning has not been well-explored in the XMC literature. In XR-Transformer, we leverage the label hierarchy defined by the HLT and train the transformer model on multi-resolution objectives.

The XMC task can be viewed as generating an 1-D image $\mathbf{y} \in \{0,1\}^L$ with binary values based on input text $\mathbf{x}$. Just like a coarsified image could be obtained by a max or mean pooling of nearby pixels,

the coarse label vector can be obtained by max-pooling of labels which are nearby in the label feature space. Once the HLT is constructed using label features, the true labels at layer $\mathbf{Y}^{(t)} \in \{0, 1\}^{N \times K_t}$ can be determined by the true labels of the child clusters at $t+1$ through a max-pooling like operation:

$$\mathbf{Y}^{(t)} = \text{binarize}(\mathbf{Y}^{(t+1)}\mathbf{C}^{(t+1)}), \tag{5}$$

and $Y_{i,\ell}^{(D)} = y_{i,\ell}$ is the original label matrix. This forms a series of learning signals with coarse-to-fine resolution and can be used to generate learning tasks with easy-to-hard objectives.

Direct use of the binarized $\mathbf{Y}^{(t)} \in \{0, 1\}^{N \times K_t}$ in Eq (5) results in information loss when merging several positive labels into one cluster. Ideally, a cluster containing several positive children is more relevant than a cluster with only one positive child. To add this lower level information to higher level learning objectives, we introduce the relevance matrix $\mathbf{R}^{(t)} \in \mathbb{R}_+^{N \times K_t}$ for layer $t$ of the XMC sub-problem where $R_{i,\ell}$ defines the non-negative important weight for $i$th instance to $\ell$th cluster. Different from cost-sensitive learning [44] for MLC (CSMLC) setting [45–47] where there is only one cost matrix explicitly derived by evaluation metrics such as F1 score, in XR-Transformer, we consider the usage of cost-sensitive learning where the relevance matrices are recursively induced by the HLT structure. Specifically, given an HLT, we recursively construct relevance matrices for $t = 1, \ldots, D$:

$$\mathbf{R}^{(t)} = \mathbf{R}^{(t+1)}\mathbf{C}^{(t+1)}, \tag{6}$$

and $\mathbf{R}^{(D)} = \mathbf{Y}^{(D)}$. Motivated by [48], we adopts the row-wise $\ell_1$ normalized relevance matrix:

$$\hat{R}_{i,j}^{(t)} = \begin{cases} \frac{R_{i,j}^{(t)}}{\|\mathbf{R}_i^{(t)}\|_1} & \text{if } Y_{i,j}^{(t)} = 1, \\ \alpha & \text{otherwise}, \end{cases}$$

where $\alpha$ is the hyper parameter to balance positive and negative weights.

**Label Shortlisting.** During training, XR-Transformer only focuses on discriminating the labels or clusters that have high chance of being positive. A necessary condition for a label at layer $t$ to be positive is that its parent label at level $t - 1$ is positive. Therefore, an intuitive approach would be to only train on the output space shortlisted by positive clusters of the parent layer. However, in practice we found this approach sometimes leads to sub-optimal result during inference with beam search. As an effort to balance explore and exploit, we further include the top-$k$ relevant clusters determined by the model learned on the parent layer to mimic the beam search during inference. Thus at layer $t$, the labels considered during training are shortlisted by the parent layer $t - 1$:

$$\mathbf{P}^{(t-1)} = \text{Top}(\mathbf{W}^{(t-1)\top}\Phi(\mathbf{X}, \Theta^{(t-1)}), k), \tag{7}$$

$$\mathbf{M}^{(t)} = \text{binarize}(\mathbf{P}^{(t-1)}\mathbf{C}^{(t)\top}) + \text{binarize}(\mathbf{Y}^{(t-1)}\mathbf{C}^{(t)\top}), \tag{8}$$

where the $\text{Top}(\cdot, k)$ operator zeros out elements in a matrix except the top-$k$ largest values in each row. For each instance $\mathbf{x}_i$, only non-zero indices of $\mathbf{M}_i$ will be included into the training objective. We can therefore define a series of learning objectives for level $t \in \{1, 2, \ldots, D\}$ as:

$$\min_{\mathbf{W}^{(t)}, \Theta} \sum_{i=1}^{N} \sum_{\ell : \mathbf{M}_{i,\ell}^{(t)} \neq 0} \hat{R}_{i,\ell}^{(t)} \mathcal{L}(Y_{i,\ell}^{(t)}, \mathbf{W}_\ell^{(t)\top}\Phi(\mathbf{x}_i, \Theta)) + \lambda\|\mathbf{W}^{(t)}\|^2, \tag{9}$$

where $\mathcal{L}$ is a point-wise loss such as hinge loss, squared hinge loss or BCE loss, $\mathbf{W}^{(t)}, \Theta$ are the model weights to be learned.

**Text Representation.** Most previous works on XMC construct text feature representation in one of two ways: statistical feature representations and semantic feature representations. Although the latter, in particular transformer models, have shown promising results on various NLP benchmarks, the self-attention mechanism makes transformers unscalable w.r.t. sequence length. To ensure efficiency, input texts are usually truncated [12, 13] which result in loss of information. On the other hand, the statistical features, such as TFIDF, are fast to construct with the whole input taken into consideration. In XR-Transformer, we use a combination of these two feature representations and each component is a complement of lost information for the other one:

$$\Phi_{cat}(\mathbf{x}, \Theta) := \left[ \frac{\Phi_{tfidf}(\mathbf{x})}{\|\Phi_{tfidf}(\mathbf{x})\|}, \frac{\Phi_{dnn}(\mathbf{x}, \Theta)}{\|\Phi_{dnn}(\mathbf{x}, \Theta)\|} \right], \tag{10}$$

---

**Algorithm 1:** Iterative_Learn$(\mathbf{X}, \mathbf{Y}, \mathbf{C}, \boldsymbol{\theta}, \mathbf{P})$

---

**Input** : $\mathbf{X}, \mathbf{Y}, \mathbf{C}, \boldsymbol{\theta}, \mathbf{P}$
$m, n \leftarrow \mathbf{C}.shape$
**if** $n = 1$ **then**
    $\mathbf{M} \leftarrow ones(N, m)$
    **if** $\boldsymbol{\theta}$ *is not fixed* **then**
        $\boldsymbol{\theta}^* \leftarrow$ Optimize (9) with $\Phi = \Phi_{dnn}$, initialize $\Theta = \boldsymbol{\theta}$
    **else**
        $\boldsymbol{\theta}^* \leftarrow \theta$
    $\mathbf{W}^* \leftarrow \operatorname{argmin}_{\mathbf{W}} \sum_{i=1}^{N} \sum_{\ell=1}^{m} \hat{R}_{i,\ell}^{(t)} \mathcal{L}(Y_{i,\ell}, \mathbf{W}_{\ell}^{\top} \Phi_{cat}(\mathbf{x}_i, \Theta)) + \lambda \|\mathbf{W}\|^2$
**else**
    $\mathbf{Y}_{prev} \leftarrow$ binarize$(\mathbf{YC})$
    $\mathbf{M} \leftarrow$ binarize$(\mathbf{PC}^{\top})$ + binarize$(\mathbf{Y}_{prev}\mathbf{C}^{\top})$
    **if** $\boldsymbol{\theta}$ *is not fixed* **then**
        $\boldsymbol{\theta}^* \leftarrow$ Optimize (9) with $\Phi = \Phi_{dnn}$ initialize $\Theta = \boldsymbol{\theta}$
    **else**
        $\boldsymbol{\theta}^* \leftarrow \boldsymbol{\theta}$
    $\mathbf{W}^* \leftarrow \operatorname{argmin}_{\mathbf{W}} \sum_{i=1}^{N} \sum_{\ell:\mathbf{M}_{i,\ell} \neq 0} \hat{R}_{i,\ell}^{(t)} \mathcal{L}(Y_{i,\ell}, \mathbf{W}_{\ell}^{\top} \Phi_{cat}(\mathbf{x}_i, \theta^*)) + \lambda \|\mathbf{W}\|^2$
**Return** : $\mathbf{W}^*, \boldsymbol{\theta}^*$

---

---

**Algorithm 2:** XR-Transformer training

---

**Input** : $\mathbf{X}, \mathbf{Y}$, pre-trained transformer $\Phi_{dnn}(\cdot, \boldsymbol{\theta}_0)$
$\hat{\mathbf{Z}}_\ell = \mathbf{v}_\ell / \|\mathbf{v}_\ell\|$; where $\mathbf{v}_\ell = \sum_{i:y_{i,\ell}=1} \Phi_{tfidf}(\mathbf{x}_i), \forall \ell \in [L]$
$\{\hat{\mathbf{C}}^{(t)}\}_{t=1}^{\hat{D}} \leftarrow$ k-means-clustering$(\hat{\mathbf{Z}})$
Generate label hierarchy $\{\hat{\mathbf{Y}}^{(t)}\}_{t=1}^{\hat{D}}$ using 5
$\boldsymbol{\theta}^* = \boldsymbol{\theta}_0, \mathbf{P} = None$
**for** $t$ *in* $1, 2, 3, \cdots, \hat{D}$ **do**
    $\hat{\mathbf{W}}, \boldsymbol{\theta}^* \leftarrow$ Iterative_Learn$(\mathbf{X}, \hat{\mathbf{Y}}^{(t)}, \hat{\mathbf{C}}^{(t)}, \boldsymbol{\theta}^*, \mathbf{P})$
    $\mathbf{P} \leftarrow$ Top$(\hat{\mathbf{W}}^{\top} \Phi_{cat}(\mathbf{X}, \boldsymbol{\theta}^*), k)$
$\mathbf{Z}_\ell = \mathbf{v}_\ell / \|\mathbf{v}_\ell\|$; where $\mathbf{v}_\ell = \sum_{i:y_{i,\ell}=1} \Phi_{cat}(\mathbf{x}_i), \forall \ell \in [L]$
$\{\mathbf{C}^{(t)}\}_{t=1}^{D} \leftarrow$ k-means-clustering$(\mathbf{Z})$
Generate label hierarchy $\{\mathbf{Y}^{(t)}\}_{t=1}^{\hat{D}}$ using 5
Fix $\boldsymbol{\theta}^*, \mathbf{P} = None$
**for** $t$ *in* $1, 2, 3, \cdots, D$ **do**
    $\mathbf{W}^{(t)}, \_ \leftarrow$ Iterative_Learn$(\mathbf{X}, \mathbf{Y}^{(t)}, \mathbf{C}^{(t)}, \boldsymbol{\theta}^*, \mathbf{P})$
    $\mathbf{P} \leftarrow$ Top$(\mathbf{W}^{(t)\top} \Phi_{cat}(\mathbf{X}, \boldsymbol{\theta}^*), k)$
**Return** : $\Phi_{cat}(\cdot, \boldsymbol{\theta}^*), \{\mathbf{C}^{(t)}\}_{t=1}^{D}, \{\mathbf{W}^{(t)}\}_{t=1}^{D}$

---

where $\Phi_{dnn}(\cdot, \Theta)$ is the transformer parametrized by $\Theta$. Once the text representation is constructed, predictions can be made by simply applying a linear projection on top of the text representation through (1).

**Training with bootstrapping.** The training of XR-Transformer consists of three steps. At first, a preliminary HLT is constructed using raw statistical features. Then a pre-trained transformer model is fine-tuned recursively from low resolution output to high resolution. At each layer $t$, fine-tuning objective (9) is optimized with initialization $\Theta = \boldsymbol{\theta}^{(t-1)*}$ the best transformer weights of layer $t - 1$. $\boldsymbol{\theta}^{(0)*}$ denotes the pre-trained transformer weights.

Unlike the transformer warmed-up with pre-trained weights, the projection weights $\mathbf{W}^{(t)}$ is trained from scratch without good initialization. At the beginning of fine-tuning, gradient flow through these cold-start (usually randomly initialized) weights will usually worsen the pre-trained components.

We leverage the recursive learning structure to tackle this issue by model bootstrapping. Concretely, $\mathbf{W}^{(t)}$ is initialized as:

$$\mathbf{W}_{init}^{(t)} := \underset{\mathbf{W}^{(t)}}{\text{argmin}} \sum_{i=1}^{N} \sum_{\ell: \mathbf{M}_{i,\ell}^{(t)} \neq 0} \hat{R}_{i,\ell}^{(t)} \mathcal{L}(Y_{i,\ell}^{(t)}, \mathbf{W}_{\ell}^{(t)\top} \Phi_{dnn}(\mathbf{x}_i, \boldsymbol{\theta}^{(t-1)*})) + \lambda \|\mathbf{W}^{(t)}\|^2, \qquad (11)$$

In practice, (11) is fast to compute since the semantic text feature for the previous layer $\Phi_{cat}(\mathbf{X}, \boldsymbol{\theta}^{(t-1)*})$ is already computed and thus (11) can be solved very quickly on CPUs with a variety of parallel linear solvers, such as LIBLINEAR [49].

Once the fine-tuning is complete, the refined HLT is constructed with the text representation that combines statistical text feature and fine-tuned semantic text embeddings. Then the ranking models are trained on top of the combined text features for the final prediction. The detailed training procedure is described in Algorithm 1 and 2.

**Inference.** The inference cost of XR-Transformer consists mainly of two parts: cost to compute transformer embedding and to retrieve relevant labels through beam search. Therefore, the inference time complexity is $O(T_{dnn} + kd\log(L))$, where $k$ is the beam size, $d$ is the concatenated feature dimension and $T_{dnn}$ is the time to compute $\Phi_{dnn}(\mathbf{x})$ for a given input. Note that even the inference is done with beam search through the refined HLT, the transformer text embedding only need to be computed once per instance.

**Connections with other tree based methods.** Although methods such as AttentionXML [8] also train on supervisions induced by label trees, the final model is a chain of sub-models which each on is learned on single-resolution. In particular, given a hierarchical label tree with depth $D$, AttentionXML will train $D$ different text encoders on each layer of the tree where as XR-Transformer trains the same transformer encoder progressively on all layers of the tree. This difference leads to a longer inference time for AttentionXML than XR-Transformer since multiple text encoders need to be queried during inference, as shown in the comparison in the inference time in Appendix A.4.2.

## 5 Experimental Results

We evaluate XR-Transformer on 6 public XMC benchmarking datasets: Eurlex-4K, Wiki10-31K, AmazonCat-13K, Wiki-500K, Amazon-670K, Amazon-3M. Data statistics are given in Table 1. For fair comparison, we use the same raw text input, sparse feature representations and same train-test split as AttentionXML [8] and other latest works [12, 13]. The evaluation metric is Precision@k (P@k), which is widely-used in XMC literature [3, 8, 12, 13, 18, 28]. The results of Propensity-score Precision@k (PSP@k) are defer to Appendix A.4.3, which focus more on tail labels' performance.

Table 1: Data statistics. $N_{train}, N_{test}$ refer to the number of instances in the training and test sets, respectively. $L$: the number of labels. $\bar{L}$: the average number of positive labels per instance. $\bar{n}$: average number of instances per label. $d_{tfidf}$: the sparse feature dimension of $\Phi_{tfidf}(\cdot)$. These six publicly available benchmark datasets, including the sparse TF-IDF features are downloaded from `https://github.com/yourh/AttentionXML` which are the same as AttentionXML [8] X-Transformer [12] and LightXML [13] for fair comparison.

| Dataset | $N_{train}$ | $N_{test}$ | $L$ | $\bar{L}$ | $\bar{n}$ | $d_{tfidf}$ |
|---|---|---|---|---|---|---|
| Eurlex-4K | 15,449 | 3,865 | 3,956 | 5.30 | 20.79 | 186,104 |
| Wiki10-31K | 14,146 | 6,616 | 30,938 | 18.64 | 8.52 | 101,938 |
| AmazonCat-13K | 1,186,239 | 306,782 | 13,330 | 5.04 | 448.57 | 203,882 |
| Wiki-500K | 1,779,881 | 769,421 | 501,070 | 4.75 | 16.86 | 2,381,304 |
| Amazon-670K | 490,449 | 153,025 | 670,091 | 5.45 | 3.99 | 135,909 |
| Amazon-3M | 1,717,899 | 742,507 | 2,812,281 | 36.04 | 22.02 | 337,067 |

**Baseline Methods.** We compare XR-Transformer with state-of-the-art (SOTA) XMC methods: AnnexML [28], DiSMEC [18], PfastreXML [41], Parabel [3], eXtremeText [24], Bonsai [50], XML-CNN [33], XR-Linear [7], AttentionXML [8], X-Transformer [12] and LightXML [13]. We obtain

Table 2: Comparison of XR-Transformer with recent XMC methods on six public datasets. Results with a trailing reference are taken from [8, Table 3] and [7, Table 3]. We obtain the results of AttentionXML*, LightXML*, X-Transformer* and XR-Transformer* on the same vectorized feature matrix provided in [8]. Due to GPU memory constraint, LightXML is not able to run on Amazon-3M. The PSP@k results are available in Appendix A.4.3.

| Methods | P@1 | P@3 | P@5 | P@1 | P@3 | P@5 | P@1 | P@3 | P@5 |
|---|---|---|---|---|---|---|---|---|---|
| | Eurlex-4K | | | Wiki10-31K | | | AmazonCat-13K | | |
| AnnexML [28] | 79.66 | 64.94 | 53.52 | 86.46 | 74.28 | 64.20 | 93.54 | 78.36 | 63.30 |
| DiSMEC [18] | 83.21 | 70.39 | 58.73 | 84.13 | 74.72 | 65.94 | 93.81 | 79.08 | 64.06 |
| PfastreXML [41] | 73.14 | 60.16 | 50.54 | 83.57 | 68.61 | 59.10 | 91.75 | 77.97 | 63.68 |
| Parabel [3] | 82.12 | 68.91 | 57.89 | 84.19 | 72.46 | 63.37 | 93.02 | 79.14 | 64.51 |
| eXtremeText [24] | 79.17 | 66.80 | 56.09 | 83.66 | 73.28 | 64.51 | 92.50 | 78.12 | 63.51 |
| Bonsai [50] | 82.30 | 69.55 | 58.35 | 84.52 | 73.76 | 64.69 | 92.98 | 79.13 | 64.46 |
| XML-CNN [33] | 75.32 | 60.14 | 49.21 | 81.41 | 66.23 | 56.11 | 93.26 | 77.06 | 61.40 |
| XR-Linear [7] | 84.14 | 72.05 | 60.67 | 85.75 | 75.79 | 66.69 | 94.64 | 79.98 | 64.79 |
| AttentionXML* | 86.93 | 74.12 | 62.16 | 87.34 | 78.18 | 69.07 | 95.84 | 82.39 | 67.32 |
| X-Transformer* | 87.61 | 75.39 | 63.05 | 88.26 | 78.51 | 69.68 | 96.48 | 83.41 | 68.19 |
| LightXML* | 87.15 | 75.95 | **63.45** | **89.67** | 79.06 | 69.87 | 96.77 | **83.98** | **68.63** |
| XR-Transformer* | **88.41** | **75.97** | 63.18 | 88.69 | **80.17** | **70.91** | **96.79** | 83.66 | 68.04 |
| | Wiki-500K | | | Amazon-670K | | | Amazon-3M | | |
| AnnexML [28] | 64.22 | 43.15 | 32.79 | 42.09 | 36.61 | 32.75 | 49.30 | 45.55 | 43.11 |
| DiSMEC [18] | 70.21 | 50.57 | 39.68 | 44.78 | 39.72 | 36.17 | 47.34 | 44.96 | 42.80 |
| PfastreXML [41] | 56.25 | 37.32 | 28.16 | 36.84 | 34.23 | 32.09 | 43.83 | 41.81 | 40.09 |
| Parabel [3] | 68.70 | 49.57 | 38.64 | 44.91 | 39.77 | 35.98 | 47.42 | 44.66 | 42.55 |
| eXtremeText [24] | 65.17 | 46.32 | 36.15 | 42.54 | 37.93 | 34.63 | 42.20 | 39.28 | 37.24 |
| Bonsai [50] | 69.26 | 49.80 | 38.83 | 45.58 | 40.39 | 36.60 | 48.45 | 45.65 | 43.49 |
| XML-CNN [33] | - | - | - | 33.41 | 30.00 | 27.42 | - | - | - |
| XR-Linear [7] | 65.59 | 46.72 | 36.46 | 43.38 | 38.40 | 34.77 | 47.40 | 44.15 | 41.87 |
| AttentionXML* | 76.74 | 58.18 | 45.95 | 47.68 | 42.70 | 38.99 | 50.86 | 48.00 | 45.82 |
| X-Transformer* | 77.09 | 57.51 | 45.28 | 48.07 | 42.96 | 39.12 | 51.20 | 47.81 | 45.07 |
| LightXML* | 77.89 | 58.98 | 45.71 | 49.32 | 44.17 | 40.25 | - | - | - |
| XR-Transformer* | **79.40** | **59.02** | **46.25** | **50.11** | **44.56** | **40.64** | **54.20** | **50.81** | **48.26** |

Table 3: Comparing training time (in hours) of DNN-based methods that produce the SOTA results in Table 2. The number following the model indicates the number of ensemble models used.

| Dataset | AttentionXML-3 | X-Transformer-9 | LightXML-3 | XR-Transformer-3 |
|---|---|---|---|---|
| Eurlex-4K | 0.9 | 7.5 | 16.9 | **0.8** |
| Wiki10-31K | **1.5** | 14.1 | 26.9 | **1.5** |
| AmazonCat-13K | 24.3 | 147.6 | 310.6 | **13.2** |
| Wiki-500K | **37.6** | 557.1 | 271.3 | 38.0 |
| Amazon-670K | 24.2 | 514.8 | 159.0 | **10.5** |
| Amazon-3M | 54.8 | 542.0 | - | **29.3** |

most baseline results from [8, Table 3] and [7, Table 3] except for the latest deep learning based algorithms [8, 12, 13]. To have fair comparison on training time, we use the same hardware (i.e., AWS p3.16xlarge) and the same inputs (i.e., raw text, vectorized features, data split) to obtain the results of AttentionXML, X-Transformer and LightXML. The hyper-parameter of XR-Transformer and more empirical results are included in Appendix A.3.

**Model Performance.** The comparisons of Precision@k (P@k) and training time are shown in Table 2 and Table 8, respectively. The proposed XR-Transformer follows AttentionXML and LightXML to use *an ensemble of 3* models, while X-Transformer uses *an ensemble of 9* models [12]. More details about the ensemble setting can be found in Appendix A.3. The proposed XR-Transformer

Table 4: Single model comparison of DNN based XMC models. Training time on p3.16xlarge with 8 Nvidia V100 GPUs $T_{train}^8$ are reported for AttentionXML, X-Transformer and XR-Transformer, whereas time on single Nvidia V100 GPU $T_{train}^1$ is reported for LightXML and XR-Transformer.

| Dataset | Method | P@1 | P@3 | P@5 | $T_{train}^1$ | $T_{train}^8$ |
|---|---|---|---|---|---|---|
| Wiki10-31K | AttentionXML-1 | 87.1 | 77.8 | 68.8 | - | **0.5** |
| | X-Transformer-1 | 87.5 | 77.2 | 67.1 | - | 3.5 |
| | LightXML-1 | 87.8 | 77.3 | 68.0 | 6.7 | - |
| | XR-Transformer-1 | **88.0** | **78.7** | **69.1** | **1.3** | **0.5** |
| Wiki-500K | AttentionXML-1 | 75.1 | 56.5 | 44.4 | - | **12.5** |
| | X-Transformer-1 | 44.8 | 40.1 | 34.6 | - | 56.0 |
| | LightXML-1 | 76.3 | 57.3 | 44.2 | 89.6 | - |
| | XR-Transformer-1 | **78.1** | **57.6** | **45.0** | **29.2** | **12.5** |
| Amazon-670K | AttentionXML-1 | 45.7 | 40.7 | 36.9 | - | 8.1 |
| | X-Transformer-1 | 44.8 | 40.1 | 34.6 | - | 56.0 |
| | LightXML-1 | 47.3 | 42.2 | 38.5 | 53.0 | - |
| | XR-Transformer-1 | **49.1** | **43.8** | **40.0** | **8.1** | **3.4** |

framework achieves new SOTA results in **14 out of 18** evaluation columns (combination of datasets and P@k), and outperforms competitive methods on the large datasets. Next, we show the training time of XR-Transformer is significantly less than other DNN-based models.

**Training Cost.** Table 8 shows the training time for these DNN-based models. To have fair comparison, all the experiments are conducted with float32 precision on AWS p3.16xlarge instance with 8 Nvidia V100 GPUs except for LightXML, which was run on single V100 GPU since multi-GPU training is not implemented. XR-Transformer consumes significantly less training time compared with other transformer based models and the shallow BiLSTM model AttentionXML. On Amazon-3M, XR-Transformer has *20x* speedup over X-Transformer while achieving even better P@k. Finally, in table 4, we compare XR-Transformer with LightXML under the single model setup (no ensemble), where XR-Transformer still consistently outperforms LightXML in P@k and training time.

Table 5: Comparing XR-Transformer with Pre-Trained and word2vec embeddings concatenated with TF-IDF features.

| Methods | P@1 | P@3 | P@5 | P@1 | P@3 | P@5 | P@1 | P@3 | P@5 |
|---|---|---|---|---|---|---|---|---|---|
| | Eurlex-4K | | | Wiki10-31K | | | AmazonCat-13K | | |
| TF-IDF | 84.14 | 72.05 | 60.97 | 85.75 | 75.79 | 66.69 | 94.64 | 79.98 | 64.79 |
| word2vec +TF-IDF | 84.35 | 71.27 | 59.10 | 86.11 | 76.92 | 66.45 | 94.53 | 79.44 | 63.94 |
| Pre-Trained +TF-IDF | 84.92 | 71.40 | 59.36 | 85.78 | 78.30 | 68.33 | 95.05 | 80.12 | 64.53 |
| XR-Transformer | **88.41** | **75.97** | **63.18** | **88.69** | **80.17** | **70.91** | **96.79** | **83.66** | **68.04** |
| | Wiki-500K | | | Amazon-670K | | | Amazon-3M | | |
| TF-IDF | 65.59 | 46.72 | 36.46 | 43.38 | 38.40 | 34.77 | 47.40 | 44.15 | 41.87 |
| word2vec +TF-IDF | 68.21 | 48.16 | 37.54 | 44.04 | 39.07 | 35.35 | 47.51 | 44.49 | 42.19 |
| Pre-Trained +TF-IDF | 70.18 | 49.82 | 38.75 | 44.55 | 38.91 | 34.77 | 49.66 | 46.41 | 43.96 |
| XR-Transformer | **79.40** | **59.02** | **46.25** | **50.11** | **44.56** | **40.64** | **54.20** | **50.81** | **48.26** |

**Comparison of Different Semantic Embeddings.** To provide more empirical justifications, that the improvement in performance comes from better semantic embedding rather than the introducing of TF-IDF features, we further tested models using Pre-Trained Transformer and word2vec embeddings concatenated with the same TF-IDF features.

Table 5 summarizes the performance of these models on all 6 datasets. In particular, word2vec is using token embedding from *word2vec-google-news-300* and for Pre-Trained we use the same setting as XR-Transformer (3-model ensemble). On large datasets such as Wiki-500K/Amazon-670K/Amazon-3M, Pre-Trained +TF-IDF has marginal improvement compared to the baseline TF-IDF features. Nevertheless, our proposed XR-Transformer still enjoy significant gain compared to Pre-Trained +TF-IDF. This suggests the major improvement is from learning more powerful neural semantic embeddings, rather than the use of TF-IDF.

**Effect of Cost-Sensitive Learning.** In Table 6, we analyze the effect of cost sensitive learning on four XMC datasets with the largest output spaces. On most datasets, cost sensitive learning via aggregated labels yields better performance than those without. We also show that cost-sensitive learning is not only beneficial to XR-Transformer, but also useful to its linear counterpart XR-Linear [7]. See Appendix A.4.1 for more results.

Table 6: Ablation of cost-sensitive learning on the single XR-Transformer model with or without Cost Sensitive (CS). Precision@1,3,5 P(@k) and Recall@1,3,5 (R@k) are reported.

| Dataset | Method | P@1 | P@3 | P@5 | R@1 | R@3 | R@5 |
|---------|--------|-----|-----|-----|-----|-----|-----|
| Wiki10-31K | XR-Transformer-1(w/o CS) | 86.8 | 77.6 | 68.8 | 5.2 | 13.6 | 19.8 |
| | XR-Transformer-1 | 88.0 | 78.7 | 69.1 | 5.3 | 13.8 | 19.9 |
| Wiki-500K | XR-Transformer-1(w/o CS) | 77.6 | 57.4 | 44.9 | 25.8 | 48.1 | 57.8 |
| | XR-Transformer-1 | 78.1 | 57.6 | 45.0 | 26.1 | 48.5 | 58.1 |
| Amazon-670K | XR-Transformer-1(w/o CS) | 49.1 | 43.8 | 40.0 | 10.3 | 25.6 | 37.7 |
| | XR-Transformer-1 | 49.0 | 43.7 | 39.9 | 10.4 | 25.7 | 37.7 |
| Amazon-3M | XR-Transformer-1(w/o CS) | 50.2 | 47.6 | 45.4 | 3.4 | 8.4 | 12.4 |
| | XR-Transformer-1 | 52.6 | 49.4 | 46.9 | 3.8 | 9.3 | 13.6 |

**Effect of Label Resolution and Text Representation.** Next, we compare the effect of label resolution on the quality of the fine-tuned transformer embeddings. We fine-tune transformer models in a non-recursive manner on a two layer HLT with different leaf cluster size. Then the fine-tuned transformer embeddings are used along or in combination with TF-IDF features to produce the predictions with refined HLT. From Figure 2 we can observe that a larger cluster size will result in worse semantic features. Figure 2 also shows that combining semantic features $\Phi_{dnn}$ with statistical features $\Phi_{tfidf}$ could in general improve the model performance.

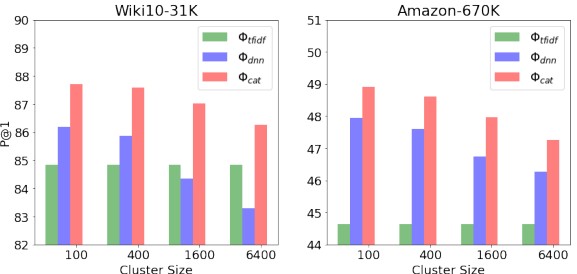

Figure 2: Comparison of BERT model fine-tuned with different label resolution. Larger cluster size means lower label resolution. Note that $\Phi_{cat}$ is the normalized concatenation of $\Phi_{tfidf}$ and $\Phi_{dnn}$.

## 6 Conclusion and Future work

In this paper, we have presented XR-Transformer approach, which is an XMC architecture that leverages multi-resolution objectives and cost sensitive learning to accelerate the fine-tuning of pre-trained transformer models. Experiments show that the proposed method establishes new state-of-the-art results on public XMC datasets while taking significantly less training time compared with earlier transformer based methods. Although the proposed architecture is designed for XMC, the ideas can be applied to other areas such as information retrieval or other DNN models such as CNNs/ResNets. Also, more extensive study is required to understand why the coarse-to-fine scheme would lead to not only faster training but better overall quality. A hypothesis is that the problem is being solved at multiple scales hence leading to more robust learning of deep transformer models.

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
