# A  Appendix

## A.1  Method Overview

At a high level sketch, the XR-Transformer model consists of three components. First, text vectorizers consists of both learnable transformer based semantic vectorizer and deterministic statistical vectorizer. Second, a recursive fine-tuning curriculum with multi-resolution output signals defined by a preliminary HLT and finally, multi-layer ranking models that gives final prediction together with a refined HLT. Figure 3 gives an illustration on the training and inference pipeline of XR-Transformer.

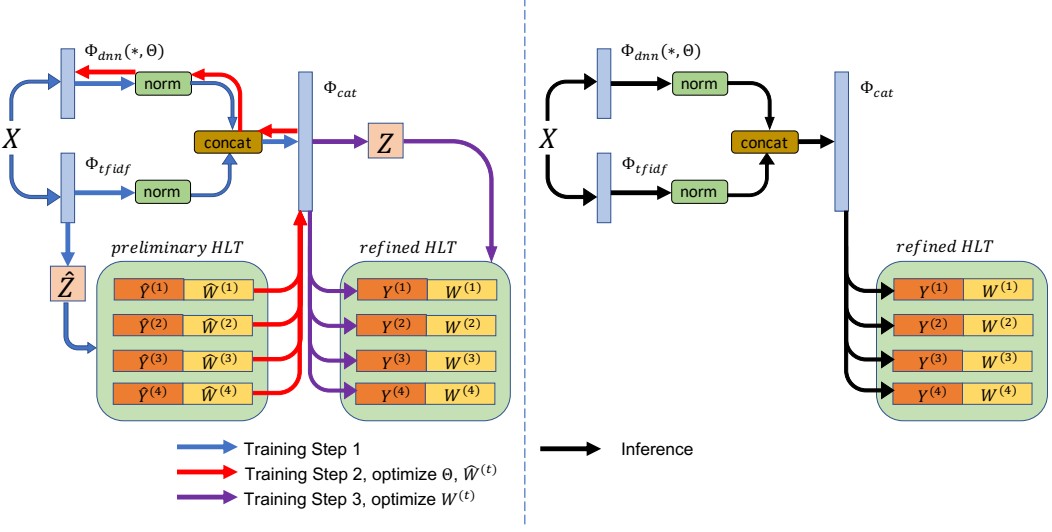

Figure 3: XR-Transformer training (left) and inference (right) architecture. XR-Transformer is trained with three steps. First, label features $\hat{\mathbf{Z}}$ are computed and is used to build preliminary hierarchical label tree (HLT) via hierarchical k-means. Then the transformer vectorizer $\Phi_{dnn}(\cdot, \Theta)$ is recursively fine-tuned on multi-resolution labels $\{\hat{\mathbf{Y}}^{(t)}\}_{t=1}^{D}$. Finally, a refined HLT is generated with $\Phi_{cat}$ and the linear ranking models $\{\mathbf{W}^{(t)}\}_{t=1}^{D}$ are learned with refined multi-resolution labels $\{\mathbf{Y}^{(t)}\}_{t=1}^{D}$. Only the transformer vectorizer $\Phi_{dnn}$, refined HLT and $\{\mathbf{W}^{(t)}\}_{t=1}^{D}$ are needed during inference.

## A.2  Evaluation Metrics

In this section, we define the evaluation metrics used in this paper. The most widely used evaluation metric for XMC is the precision at k (Prec@k) and recall at k (Recall@k), which are defined as:

$$\text{Prec@k} = \frac{1}{k} \sum_{l=1}^{k} y_{rank(l)} \tag{12}$$

$$\text{Recall@k} = \frac{1}{\sum_{i=1}^{L} y_i} \sum_{l=1}^{k} y_{rank(l)} \tag{13}$$

where $y \in \{0,1\}^L$ is the ground truth label and $rank(l)$ is the index of the $l$-th highest predicted label.

For performance comparison on tail labels, we also report propensity scored precision (PSP@k), which is defined as:

$$\text{PSP@k} = \frac{1}{k} \sum_{l=1}^{k} \frac{y_{rank(l)}}{p_{rank(l)}} \tag{14}$$

where $p_{rank(l)}$ is the propensity score at position $rank(l)$ [41]. The metric involves application specific parameters $A$ and $B$. For consistency, we use the same setting as AttentionXML [8] for all datasets.

## A.3   Model Hyperparameters

For each sub-XMC problem, XR-Transformer fine-tunes the transformer encoder with Adam [51] with linear learning rate schedule and a batch size of 32 per GPU (256 total). The dropout probability is set to be 0.1 for all models and bootstrapping (11) is used for all fine-tuning problems except for the root layer. The detailed hyperparameters are listed in Table 7.

### A.3.1   Model Ensemble Setup

For AttentionXML, LightXML and X-Transformer, we use the same ensemble setting provided in the paper. In particular, AttentionXML uses ensemble of 3 models, LightXML uses ensemble of 3 transformer encoders and X-Transformer uses 9 model ensemble with BERT, RoBerta, XLNet large models with three difference indexers. For our method, we follow the setting of LightXML and use the ensemble of BERT, RoBerta and XLNet for Eurlex-4K, Wiki10-31K and AmazonCat-13K, and ensemble of three BERT model for Wiki-500K, Amazon-670K and Amazon-3M.

Table 7: Hyperparameters for XR-Transformer. $HLT_{prelim}$ and $HLT_{refine}$ define the structures of the preliminary and refined hierarchical label trees. $lr_{max}$ is the maximum learning rate used in fine-tuning. $n_{step}$ is the total number of optimization steps across the multi-resolution fine-tuning. $N_x$ is the number of input tokens after text truncation. $\alpha$ is the hyper-parameter for cost sensitive learning. $\lambda$ is the weight for the regularization term.

| Dataset | $HLT_{prelim}$ | $HLT_{refine}$ | $lr_{max}$ | $n_{step}$ | $N_x$ | $\alpha$ | $\lambda$ |
|---|---|---|---|---|---|---|---|
| Eurlex-4K | 16-256-3956 | 4-32-256-3956 | $5 \times 10^{-5}$ | 2400 | 128 | 1.0 | 0.5 |
| Wiki10-31K | 128-2048-30938 | 8-128-2048-30938 | $1 \times 10^{-4}$ | 4000 | 256 | 0.25 | 0.25 |
| AmazonCat-13K | 128-1024-13330 | 128-1024-13330 | $1 \times 10^{-4}$ | 45000 | 256 | 2.0 | 2.0 |
| Wiki-500K | 64-512-4096-32768 | 8-256-8192-501070 | $1 \times 10^{-4}$ | 60000 | 128 | 0.25 | 0.25 |
| Amazon-670K | 128-2048-32768 | 8-256-8192-670091 | $1 \times 10^{-4}$ | 20000 | 128 | – | 1.0 |
| Amazon-3M | 128-2048-32768 | 8-256-8192-2812281 | $1 \times 10^{-4}$ | 30000 | 128 | 0.125 | 0.125 |

## A.4   More Empirical Results

Table 8: Comparing training time (in hours) of DNN-based methods that produce the SOTA results in Table 2. The number following the model indicates the number of ensemble models used.

| Dataset | AttentionXML-3 | X-Transformer-9 | LightXML-3 | XR-Transformer-3 |
|---|---|---|---|---|
| Eurlex-4K | 0.9 | 7.5 | 16.9 | **0.8** |
| Wiki10-31K | **1.5** | 14.1 | 26.9 | **1.5** |
| AmazonCat-13K | 24.3 | 147.6 | 310.6 | **13.2** |
| Wiki-500K | **37.6** | 557.1 | 271.3 | 38.0 |
| Amazon-670K | 24.2 | 514.8 | 159.0 | **10.5** |
| Amazon-3M | 54.8 | 542.0 | - | **29.3** |

### A.4.1   Cost-sensitive Learning

In table 9, we show that the cost-sensitive learning via recursive label aggregation is beneficial to both the linear XR-Linear and the Transformer-based XR-Transformer model. On most datasets, cost sensitive learning yields better performance.

### A.4.2   Inference Speed

We report the inference time for AttentionXML, X-Transformer, LightXML and XR-Transformer on XMC datasets in table 10. The inference results (millisecond per sample) are evaluated on single GPU and single CPU for most comparing models except for AttentionXML, which is evaluated with multi-GPUs on Wiki-500K, Amazon-670K and Amazon-3M. AttentionXML requires model-parallelism on those largest datasets otherwise it may be out-of-memory on a single-GPU setup.

Table 9: Ablation of cost-sensitive learning on the single XR-Linear and XR-Transformer model with or without Cost Sensitive (CS). Precision@1,3,5 (P@k) and Recall@1,3,5 (R@k) are reported.

| Dataset | Method | P@1 | P@3 | P@5 | R@1 | R@3 | R@5 |
|---------|--------|-----|-----|-----|-----|-----|-----|
| Wiki10-31K | XR-Linear-1(w/o CS) | 84.5 | 73.1 | 64.1 | 5.0 | 12.8 | 18.4 |
| | XR-Linear-1 | 85.7 | 75.0 | 65.2 | 5.1 | 13.1 | 18.7 |
| | XR-Transformer-1(w/o CS) | 86.8 | 77.6 | 68.8 | 5.2 | 13.6 | 19.8 |
| | XR-Transformer-1 | 88.0 | 78.7 | 69.1 | 5.3 | 13.8 | 19.9 |
| Wiki-500K | XR-Linear-1(w/o CS) | 66.7 | 47.8 | 37.3 | 21.7 | 39.5 | 47.6 |
| | XR-Linear-1 | 67.7 | 48.4 | 37.6 | 22.2 | 40.4 | 48.6 |
| | XR-Transformer-1(w/o CS) | 77.6 | 57.4 | 44.9 | 25.8 | 48.1 | 57.8 |
| | XR-Transformer-1 | 78.1 | 57.6 | 45.0 | 26.1 | 48.5 | 58.1 |
| Amazon-670K | XR-Linear-1(w/o CS) | 44.0 | 39.2 | 35.4 | 9.2 | 22.7 | 33.2 |
| | XR-Linear-1 | 44.4 | 39.3 | 35.6 | 9.4 | 23.0 | 33.6 |
| | XR-Transformer-1(w/o CS) | 49.1 | 43.8 | 40.0 | 10.3 | 25.6 | 37.7 |
| | XR-Transformer-1 | 49.0 | 43.7 | 39.9 | 10.4 | 25.7 | 37.7 |
| Amazon-3M | XR-Linear-1(w/o CS) | 46.8 | 44.0 | 41.9 | 2.9 | 7.3 | 10.7 |
| | XR-Linear-1 | 50.1 | 46.6 | 44.0 | 3.4 | 8.4 | 12.2 |
| | XR-Transformer-1(w/o CS) | 50.2 | 47.6 | 45.4 | 3.4 | 8.4 | 12.4 |
| | XR-Transformer-1 | 52.6 | 49.4 | 46.9 | 3.8 | 9.3 | 13.6 |

Table 10: Comparison of XR-Transformer with recent XMC methods on public datasets w.r.t. inference time. Times are recorded with single Nvidia V100 GPU and batch size of 1 except for the numbers with superscript $^*$, where model parallel was used and inference was done with 8 GPUs. The unit is milliseconds per sample.

| Dataset | AttentionXML-1 | X-Transformer-1 | LightXML-1 | XR-Transformer-1 |
|---------|---------------|-----------------|------------|------------------|
| Eurlex-4K | 12.7 | 48.2 | 24.7 | 22.3 |
| Wiki10-31K | 20.0 | 48.1 | 27.1 | 39.1 |
| AmazonCat-13K | 14.4 | 47.6 | 24.1 | 26.1 |
| Wiki-500K | 80.1$^*$ | 48.1 | 27.3 | 33.9 |
| Amazon-670K | 76.0$^*$ | 48.0 | 23.3 | 30.9 |
| Amazon-3M | 130.5$^*$ | 50.2 | - | 35.2 |

### A.4.3 Propensity-score Precision

We report the propensity scored precision (PSP@k) metric on large scale XMC datasets for PfastreXML, Parabel, AttentionXML and XR-Transformer in Table 11. For consistency, we use the same setting as AttentionXML [8] for all datasets.

Table 11: Comparison of XR-Transformer with recent XMC methods on 3 large public datasets w.r.t. PSP@k (propensity scored precision at $k = 1, 3, 5$). Results with a trailing reference are taken from [8, Table 6]. We obtain the results of AttentionXML$^*$ and XR-Transformer on the same vectorized feature matrix provided in [8] and same hyper-parameter for propensity score calculation.

| | Wiki-500K | | | Amazon-670K | | | Amazon-3M | | |
|---------|-------|-------|-------|-------|-------|-------|-------|-------|-------|
| Methods | PSP@1 | PSP@3 | PSP@5 | PSP@1 | PSP@3 | PSP@5 | PSP@1 | PSP@3 | PSP@5 |
| PfastreXML [41] | 32.02 | 29.75 | 30.19 | 20.30 | 30.80 | 32.43 | **21.38** | 23.22 | 24.52 |
| Parabel [3] | 26.88 | 31.96 | 35.26 | 26.36 | 29.95 | 33.17 | 12.80 | 15.50 | 17.55 |
| AttentionXML$^*$ | 30.69 | 38.92 | 44.00 | 30.25 | 33.88 | 37.18 | 15.42 | 18.32 | 20.48 |
| XR-Transformer$^*$ | **35.76** | **42.22** | **46.36** | **36.16** | **38.39** | **40.99** | 20.52 | **23.64** | **25.79** |

### A.5 Potential Negative Societal Impacts

This paper focus on the acceleration of the training algorithms on XMC. To the best of our knowledge, our work poses no negative societal impacts.