# OpenReview forum: "Fast Multi-Resolution Transformer Fine-tuning for Extreme Multi-label Text Classification"
_NeurIPS.cc/2021/Conference — NeurIPS 2021 Poster_

### Official Review · Reviewer_ihmk · 2021-07-06

**Rating:** 6
**Confidence:** 4

**Summary:**

To accelerate the fine-tuning procedure of transformer models on XMC, the paper proposes XR-Transformer, which is recursively fine-tuned on coarse-to-fine multi-resolution tasks and cost sensitive learning. Experiments on the benchmarks show that XR-Transformer generally outperforms previous methods and is faster than the competing transformer-based models.

**Limitations And Societal Impact:**

 The paper should not pose any negative societal impacts.

**Main Review:**

The paper proposes a novel method to accelerate fine-tuning transformer-based models for XMC, which is inspired by the multi-resolution learning in image generation and curriculum learning. This technique allows for fast fine-tuning and good performance. Whereas concatenating neural features and TF-IDF features to produce the predictions is already discussed in the X-Transformer paper and thus is not new.

The paper is technically sound and well written overall. The method is generally well established. The experiment results seem to be solid. The DNN-based methods use the same inputs for fair comparisons, and the running time is also well reported. Since the results of the DNN-based methods are slightly different from the original X-Transformer and LightXML papers, it could be better to give the number of run times, the standard deviation of the results, and the statistical significance. In addition, there is a slight gap between the training time of ensemble model (Table 3) and single model (Table 4), this might be improved by adding the standard deviation.

 **Question for the authors:**

1. The text inputs are truncated to 128 tokens, while the documents are generally longer on average. As the training time is significantly reduced, perhaps XR-Transformer could use longer input text to obtain even better results?
2. The authors mention the PSP and tail labels, do the coarse and fine resolutions have connections with the label frequency? Such as higher resolutions focus more on tail labels. Are the HLT and indexing matrices $C$ related to the tail labels? I would like to see more discussion about this.

**Typo:**

- Line 66  model -> models
- Line 247 The **comparison** of ... **are** ...

**Time Spent Reviewing:**

4.5

---

> ### Author Response · Authors · 2021-08-10
> **Response to Reviewer ihmk**
>
> We thank reviewer ihmk for the suggestions and we address the specific comments below.
>
>
> > The number of run times, the standard deviation of the results, and the statistical significance.
>
> * Thanks for the suggestion. We conducted t-test on the P@1 score on predictions of `XR-Transformer` and `LightXML` on individual test instances and got p-values of 0.015, 0.006 and 0.0002 for Eurlex-4K, Wiki10-31K and AmazonCat-13K respectively, indicating significant difference between the two model predictions. We will include the standard deviation and statistical significance of the results for multiple runtimes in the final version.
>
>
> > Can the result be improved by using less aggressive text truncation?
>
> * Thanks for your suggestion, we believe it is possible that the result could be further improved by using longer text inputs. In the paper, we follow the setting used in `X-Transformer` to truncate text input at 128 tokens. We will include experiments on longer input text in the final version.
>
> > Do the coarse and fine resolutions have connections with the label frequency?
>
> * In most XMC tasks, the original label signal has a long tail distribution, meaning that the tail labels will be included only in very few training steps. However, by looking at coarse resolution label signals, tail labels will receive more updates since the signals of the other labels within the same cluster will be shared. That being said, transformer fine-tuned from coarse-to-fine signals will have more chance to see tail labels compared with those fine-tuned on fine grained signals only. The effect on the final model performance, however, could be an interesting direction for follow up studies in this direction.

---

### Official Review · Reviewer_HoYN · 2021-07-12

**Rating:** 7
**Confidence:** 4

**Summary:**

This paper proposes a coarse-to-fine learning framework to solve the problem of extreme multi-label text classification.  Experiments show the model not only accelerates existing models 20x times but also achieves a slightly better performance on the dataset with an extremely large label set. The techniques are sound and the empirical results are promising.

**Limitations And Societal Impact:**

1. The first figure in the Appendix that overviews the model architecture is relatively hard to understand. It would be very helpful for readers to understand this paper if the model architecture is presented in a more understandable way.
2. To the best of my knowledge, this paper has no potential negative societal impact.
3. The authors do not address the limitations mentioned in Sec. 6 (understand why the coarse-to-fine scheme leads to better performance). The experiments discussed in above weaknesses 1 may help them better understand where the improvements come from.

**Main Review:**

Advantages:
1. The paper is well-written and well-organized. The authors include enough background information and clearly illustrate the efficiency-resolution trade-off in this task. Readers without prior knowledge of the XMC problem can easily follow it.
2. The paper is well-motivated. The proposed method is natural, sound, and potentially helpful in other problems.
3. The model achieves state-of-the-art performance on 14 out of 18 metrics on 6 datasets.
4. The authors will publish their codes.
5. Though the empirical improvements over the baseline models are not very significant on many datasets, the great time efficiency makes the model more applicable in many real-world scenarios.


Weaknesses:
1. Since the proposed method incorporates TF-IDF into the text representation and incorporating TF-IDF is often helpful in improving model performance, these improvements may mainly come from TF-IDF instead of the proposed framework. The superiority of the proposed framework would be more convincing if the authors enhance baselines models with TF-IDF and compare the proposed XR-transformer with the enhanced baselines OR compare the unenhanced-XR-transformer with baseline models.


**Time Spent Reviewing:**

3

---

> ### Author Response · Authors · 2021-08-10
> **Response to Reviewer HoYN**
>
> We thank reviewer HoYN for the suggestions and we address the specific comments below.
>
> > Is the improvement coming from TFIDF?
>
> * In table 2, we compare `XR-Transformer` with `X-Transformer` and `XR-Linear`, where all of these methods are using the same set of TFIDF features. We show that either using only statistical TFIDF features (`XR-Linear`) or using TFIDF features with single-resolutional-tuned Transformers (`X-Transformer`) results in worse performance than `XR-Transformer`. We further compare the *unenhanced-XR-transformer* where the instance embeddings are from pre-trained Transformer models (Bert, Roberta, XLNet) or word2vec embeddings (`word2vec-google-news-300`). The results (see common reply) indicate that the improvement is mainly from the fine-tuned instance embedding rather than introduction of TFIDF features.
>
>
> > Present model architecture.
> * In the appendix we present Fig 3 which illustrates the model architecture for the propose method. We’ll add a pointer from main text pointing to Fig 3.

---

### Official Review · Reviewer_9rJq · 2021-07-16

**Rating:** 7
**Confidence:** 4

**Summary:**

The authors address the problem of fine-tuning transformers models for extreme multi-label text classification (XMLTC). Even with pre-trained transformer models, fine-tuning it to the task of predicting for large label space is computationally expensive and requires significant GPU resources. The authors propose a new routine to train the X-Transformers-like model that significantly reduces that computational cost. The routing resembles AttentionXML learning routine, where one model is trained by tree level to predict meta-labels at that level. Training starts from the top level of the tree, and the model for the next level is initialized with weights of the final model from the previous level. Here underlying transformer model is first tuned on the task of predicting the same meta-labels. It starts with tuning using meta-labels from the top level and then continues tuning layer by layer. In addition, the authors apply a few additional techniques like cost-sensitive training (positive meta-labels are weighted by a number of relevant labels they consist) and bootstrapping of output weights for meta-labels/labels based on current transformers embeddings. After tuning, new label-partitioning is constructed, and the model is retrained with fixed transformer weights. The authors evaluate the attractiveness of their method on six popular benchmark datasets. New routing leads to a significant reduction in training time at the same often achieving better predictive performance.

**Ethical Concerns:**

No ethical issues were found.

**Limitations And Societal Impact:**

No issues with negative social impact were found.

**Main Review:**

Strengths:
+ The organization of this paper is good, and it's easy to understand.
+ The paper has clear motivation.
+ Proposed method significantly reduces the computational cost of training achieving better or similar results to its parent method X-Transformers and SOTA LightXML and AttentionXML.
+ Reduction of training cost makes XMLC more accessible cost, which is vital for further research and practitioners deploying such models.
+ Proposed training routine should be applicable to other label tree-based approaches methods with transformers and possibly other types of underlining deep models.
+ Solid experimental evaluation that proves the attractiveness of the proposed method.
+ The appendix seems to consist of enough details to reproduce this work and results.

Weaknesses:
- I don't find any serious issue in this work. It seems to me that the scope of contribution is a bit limited since the proposed approach is composed of elements that were already explored by the XC community, but this is a very successful combination.

Other comments:
* "Nevertheless, multi-resolution learning has not been well-explored in the XMC literature. In XR-Transformer, we leverage the label hierarchy defined by the HLT and train the transformer model on multi-resolution objectives." - I would argue with that statement, as I mentioned in summary, your routine resembles AttentionXML approach. And while it's not directly multi-resolution learning, I would personally acknowledge it here more.
* After reading the appendix (Table 6 to be precise), I realized that the initial Hierarchical Label Tree (HLT) used for tuning is different from the refined HLT (I mean pre-leaf structure). The initial is more "wide", which makes sense since you want to have more fine-grained meta-labels. But I don't find this stated in the main text. Maybe it's worth commenting on it.
* I think that Recursive_Learn procedure could be a bit more readable and general if presented in an iterative fashion, where the whole HLT is already given.
* There is a mistake in Table 4, Row "Wiki-500, X-Transformers-1" is the same as the row "Amazon-670K, X-Transformers-1".

---

Edit after the authors' response: I think the authors responded well to the reviewers' comments. I would like to keep my rating and recommend this paper for acceptance.

**Time Spent Reviewing:**

5

---

> ### Author Response · Authors · 2021-08-10
> **Response to Reviewer 9rJq**
>
> We thank reviewer 9rJq for the suggestions and we address the specific comments below.
>
> > AttentionXML is multi-resolution learning.
>
> * Yes, we acknowledge that `AttentionXML` also adopts training on multi-resolutional supervisions. However, we want to point out the differences between our proposed method and `AttentionXML`. Given a hierarchical label tree with depth $d$, `AttentionXML` will train $d$ different text encoders on each layer of the tree where as `XR-Transformer` will train the same encoder progressively on all $d$ layers of the tree. This difference leads to a longer inference time for `AttentionXML` than `XR-Transformer` when multiple text encoders need to be queried during inference (*Table 9, Appendix A.4.3*). We will add discussion on the similarities and differences of these two approaches in the final version.
>
>
> > The structural difference between preliminary HLT and refined HLT.
>
> * The motivation behind using a preliminary HLT with *wider* structure is that we would start fine-tuning text encoder on a reasonable level of label resolution instead of only 4 or 8 labels, where supervision signals would be too coarse. Thanks for the comment and we will add discussion in the main text.
>
> > Iterative fashion rather than the recursive fashion.
>
> * Thanks for the suggestion, we will add discussion for the method where the algorithm is presented in an iterative fashion.
>
> > Mistake at table 4.
>
> * Thanks for pointing this out. The `X-Transformer-1` model for `Wiki-500K` has P@1,3,5=76.05, 55.73, 43.46 and 54hr training time with 8 GPUs. We will correct the mistake in the final version.

---

> > ### Comment · Reviewer_9rJq · 2021-08-24
> > **Response to the authors' response**
> >
> > Dear authors,
> >
> > thank you for addressing my and other reviewers' comments, your responses strengthen my opinion, that your work should be accepted at the conference.
> >
> > Additional note: while the proposed approach significantly reduces the computational cost of training compared to X-Transformers, it is still expensive for larger that sets. That's why I believe it would be beneficial for the XC community to not only make the code available publicly but also the final parameters of the underling transformer of XR-Transformers presented in the paper.

---

> > > ### Author Response · Authors · 2021-08-31
> > > **Thanks for your response**
> > >
> > > Thanks for your response. We will make both codes/parameters/models publicly available.

---

### Official Review · Reviewer_YuK3 · 2021-07-20

**Rating:** 5
**Confidence:** 4

**Summary:**

The authors propose an approach for extreme multi-label classification  that is based on a transformer network and leverages a reduction approach that breaks down the problem to smaller ones via a hierarchical approach that constructs a label tree. The authors use both the representation of the transformer as well as classical TF-IDF ones as the final one for the model. The approach is validated in 6 large datasets and compared with other state-of-the-art approaches.

**Limitations And Societal Impact:**

Still the approach is expensive and to be honest my take away if that TF-IDF is still a powerful scheme.

No negative societal impact.

**Main Review:**

Using tree structures for extreme multilabel classification is not new as an approach. The authors can refer to approaches like the ones developed by Langford and team (strangely enough these works are not cited). So, reductions are not new.

What it seems to be new is the multi-resolution learning. The authors have some results on the size of the clusters but is not very clear how much this way of training helps. I think what is missing here is a deeper exploration in terms of experiments. For, example is interesting that the TF-IDF features alone do not show variation based on the cluster size while DNN ones yes. The authors should put some more effort here to better understand why this happens.

Improvements are rather small, even though the proposed approach improves in most of the cases. Training time is greatly improved though.
We need to be in a really large output space to see a good difference in the metrics. It would be nice if the authors could elaborate a bit more on this.

Would it be possible also to have the results of Table 2 with/wo TF-IDF features?

Why the authors did not try to compare with simple concatenations of TF-IDF and aggregations based on a pre-trained language model or even a word2vec approach.

Is after-all the use of the simple statistical features that helps to improve the performance?

**Time Spent Reviewing:**

2.5

---

> ### Author Response · Authors · 2021-08-10
> **Response to Reviewer YuK3**
>
> We thank reviewer YuK3 for the suggestions and we address the specific comments below.
>
> > Tree structure is not new.
>
> * Thanks for the reference, we will add more discussion on prior works using tree structures and add a citation to Langford et. al.
>
> > In Fig (2), the TF-IDF features alone do not show variation based on the cluster size while DNN ones do.
>
> * The $\phi_{tfidf}$​ is used as a baseline in Fig (2) as the model does not utilize the fine-tuned embeddings. Therefore the scores of $\phi_{tfidf}$ will be independent of how the embeddings are fine-tuned.
>
> > Improvements are rather small. Only have good difference on very large output space.
>
> * It is true that the improvement on model accuracies are more obvious on very large output space. For datasets with small output size, fine-tuning Transformer models would be easier even on full label resolution, and therefore the improvement of multi-resolutional training is not as obvious as those with large output spaces.
> * To achieve good performance, methods like `LightXML` train on high-resolutional signals (all labels for Eurlex-4K, Wiki10-31K, AmazonCat-13K, ~10k labels for Wiki-500K/Amazon-670K) at the price of very long training time. For the same reason, the method cannot scale to 3M output space for Amazon-3M. For `XR-Transformer`, however, we can fine-tune the Transformer models with much less training time through multi-resolutional training and produce same or better quality input embeddings and can easily scale to very large output spaces.
>
> > Simple TFIDF concatenation with pre-trained model even word2vec approach.
>
> * In the common reply to all reviewers we included the comparison on 6 XMC datasets using TFIDF concatenation with pre-trained transformer and word2vec approach under same setting as XR-Transformer. The only differences between `Word2Vec+TFIDF`, `PreTraind+TFIDF` and `XR-Transformer` are the instance embeddings. Simply using the TFIDF concatenation with the pre-trained transformer embeddings or word2vec embeddings seems to only have marginal improvement over TFIDF baseline and is much worse than using `XR-Transformer` fine-tuned embeddings.
>
> > Is the improvement coming from TFIDF?
>
> * In terms of improvement on training time, the fine-tuning of Transformers dominates the training time and gain is mainly from the proposed recursive fine-tuning scheme.
> * In terms of accuracy, in table 2, we compare `XR-Transformer` with `X-Transformer` and `XR-Linear`, where all of these methods are using the same set of TFIDF features. We show that either using only statistical TFIDF features (`XR-Linear`) or using TFIDF features with single-resolutional-tuned Transformers (`X-Transformer`) results in worse performance than `XR-Transformer`. We further provide the comparison with concatenating TFIDF with (1) pre-trained transformer embeddings (2) word2vec embeddings (See common reply). The results on this additional experiment also suggest that the improvement is mainly coming from neural embeddings rather than TFIDF.
> * Therefore, we believe that the improvement is from properly fine-tuning the transformer rather than simply using TFIDF features.

---

> > ### Author Response · Authors · 2021-08-31
> > **Any further comments?**
> >
> > We really appreciate your valuable comments. Since the discussion period is ending soon, we would like to hear your thoughts after reviewing our additional experiments in https://openreview.net/forum?id=gjBz22V93a&noteId=kqon0pfsmnx ?
> >
> > Sincerely,
> >
> > Authors of this submission

---

### Author Response · Authors · 2021-08-10
**To all Reviewers:**

We thank all the reviewers very much for their valuable comments and constructive suggestions to strengthen our work. In addition to the response to specific reviewers, here we address the common questions raised by Reviewer YuK3 and HoYN.

> Is the improvement coming from using TFIDF features? How’s the performance if simple pre-trained model or word2vec embeddings are used?

 * In Table 2 of the paper, we compare `XR-Transformer` with `X-Transformer` and `XR-Linear`, where all of these methods are using the same set of TFIDF features. We show that either using only statistical TFIDF features (`XR-Linear`) or using TFIDF features with single-resolutional-tuned Transformers (`X-Transformer`) results in worse performance than `XR-Transformer`.
 * To provide more empirical justifications, we further tested models using pre-trained Transformer and `word2vec` embeddings concatenated with the same TFIDF features. The following table summarizes the performance of these models on all 6 datasets. In particular, `word2vec` is using token embedding from `word2vec-google-news-300` and for PreTrained we use the same setting as `XR-Transformer` (3-model ensemble of Bert, Roberta, XLNet). On large datasets such as *wiki-500k*/*amazon-670k*/*amazon-3m*, `PreTrained+TFIDF` has marginal improvement compared to the baseline TFIDF features. Nevertheless, our proposed `XR-Transformer` still enjoy significant gain compared to `Pretrained+TFIDF`. This suggests the major improvement is from learning more powerful neural embeddings, rather than the use of TFIDF.

| Dataset       | Method          | P@1   | P@3   | P@5   |
|---------------|-----------------|-------|-------|-------|
| Eurlex-4K     | TFIDF Only      | 84.14 | 72.05 | 60.97 |
|               | Word2Vec+TFIDF  | 84.35 | 71.27 | 59.10 |
|               | PreTraind+TFIDF | 84.92 | 71.40 | 59.36 |
|               | XR-Transformer  | **88.31** | **75.96** | **63.03** |
| Wiki10-31K    | TFIDF Only      | 85.75 | 75.79 | 66.69 |
|               | Word2Vec+TFIDF  | 86.11 | 76.92 | 66.45 |
|               | PreTraind+TFIDF | 85.78 | 78.30 | 68.33 |
|               | XR-Transformer  | **88.63** | **80.20** | **70.95** |
| AmazonCat-13K | TFIDF Only      | 94.64 | 79.98 | 64.79 |
|               | Word2Vec+TFIDF  | 94.53 | 79.44 | 63.94 |
|               | PreTraind+TFIDF | 95.05 | 80.12 | 64.53 |
|               | XR-Transformer  | **96.84** | **84.26** | **68.70** |
| Wiki-500K     | TFIDF Only      | 65.59 | 46.72 | 36.46 |
|               | Word2Vec+TFIDF  | 68.21 | 48.16 | 37.54 |
|               | PreTraind+TFIDF | 70.18 | 49.82 | 38.75 |
|               | XR-Transformer  | **78.38** | **57.90** | **44.28** |
| Amazon-670K   | TFIDF Only      | 43.38 | 38.40 | 34.77 |
|               | Word2Vec+TFIDF  | 44.04 | 39.07 | 35.35 |
|               | PreTraind+TFIDF | 44.55 | 38.91 | 34.77 |
|               | XR-Transformer  | **49.85** | **44.37** | **40.49** |
| Amazon-3M     | TFIDF Only      | 47.40 | 44.15 | 41.87 |
|               | Word2Vec+TFIDF  | 47.51 | 44.49 | 42.19 |
|               | PreTraind+TFIDF | 49.66 | 46.41 | 43.96 |
|               | XR-Transformer  | **54.04** | **50.64** | **48.10** |

---

### Decision · Program_Chairs · 2021-09-27

**Decision:**

Accept (Poster)

**Comment:**

The paper introduces an improved version of a transformer-based algorithm for extreme multi-label classification. The empirical results show significant speed-ups and very competitive predictive performance. The paper is well-written and the proposed modifications are sound. The reviewers were satisfied by the responses given by the authors.